# Free Final-Time Fuel-Optimal Powered Landing Guidance Algorithm Combing Lossless Convex Optimization with Deep Neural Network Predictor

**Wenbo Li [1] and Shengping Gong [2],***

1    School of Aerospace Engineering, Tsinghua University, Beijing 100084, China; liwb21@mails.tsinghua.edu.cn
2    School of Astronautics, Beihang University, Beijing 100191, China
*    Correspondence: gongsp@buaa.edu.cn

**Abstract:** The real-time guidance algorithm is the key technology of the powered landing. Given the lack of real-time performance of the convex optimization algorithm with free final time, a lossless convex optimization (LCvx) algorithm based on the deep neural network (DNN) predictor is proposed. Firstly, the DNN predictor is built to map the optimal final time. Then, the LCvx algorithm is used to solve the problem of fuel-optimal powered landing with the given final time. The optimality and real-time performance of the proposed algorithm are verified by numerical examples. Finally, a closed-loop simulation framework is constructed, and the accuracy of landing under various disturbances is verified. The proposed method does not need complex iterative operations compared with the traditional algorithm with free final time. Therefore, the computational efficiency can be improved by an order of magnitude.

**Keywords:** powered landing guidance; lossless convex optimization; successive convex optimization; deep neural network; model predictive control

## 1. Introduction

With the development of aerospace technology, the demand for landings to explore the moon and other exoplanets has increased dramatically. The future mission requires the vehicle to achieve a high-precision landing to explore regions of scientific value efficiently [1]. In recent years, rocket vertical recovery technology developed from planetary soft-landing technology has attracted the attention of many researchers. The purpose of the technology is to guide the rocket core stage to achieve vertical recovery, which significantly reduces the cost of space transportation [2]. The above scenarios both lead to new requirements for the powered landing guidance algorithm. Due to the complex dynamic environment, the initial state of the powered landing phase cannot be accurately predicted in advance, so the tracking guidance method based on offline calculation cannot be adopted. Therefore, the real-time optimal guidance algorithm has become the key technology for the powered landing problem. The primary purpose of the work is to design a high-precision landing guidance algorithm based on convex optimization fusing the deep neural network (DNN) predictor, which satisfies optimality and real-time performance.

The early guidance algorithms mainly adopted analytical algorithms, including the gravity turn [3,4] and Apollo polynomial guidance algorithm [5]. The gravity turn guidance algorithm cannot guarantee the landing accuracy. The Apollo guidance algorithm can achieve a high-precision lunar landing. However, the algorithm cannot deal with process constraints or guarantee the optimality of fuel consumption [6]. After the 1960s, many scholars have proposed various indirect guidance methods based on Pontriagin's minimum principle. These methods transform the optimal control problem into a two-point boundary value problem (TPBVP), and different methods were used to solve the TPBVP [7–10]. However, the convergence of the indirect method is sensitive to the costate variables, and

how to select the initial costate variables needs further research. To sum up, the traditional guidance algorithms lack real-time and optimality performance, and it is difficult to achieve an accurate landing in a complex dynamic environment.

In recent years, many scholars have proposed some potential real-time guidance algorithms, which can be divided into two categories. The first category is based on convex optimization (CVX) [11–17]. Açikmeşe [11] and Blackmore [12] first used variable substitution and lossless convex optimization (LCvx) to deal with the nonconvex dynamic equations and constraints, and the second-order cone programming (SOCP) was built to solve the Mars landing problem. Liu and Lu [13,14] proposed the principle of successive convex optimization (SCvx). By linearizing the constraint and adding the trust-region constraint, the nonlinear optimal control problem was transformed into a sequence of sub-SOCPs. This method can deal with more complex dynamic constraints and has been successfully applied to various aerospace guidance problems, including powered landing. The most remarkable advantage of the CVX-based algorithm is that it has strict convergence proof, and the algorithm can obtain the optimal solution in polynomial time. However, due to the limitation of the current onboard CPU, it is still hard to achieve real-time guidance. The second category is ML-based algorithms [18–22]. Sánchez [19] proposed a supervised learning guidance algorithm based on DNN. Firstly, the indirect method generated abundant training samples, and then the DNN was used to learn the relationship between state and control variables. The control signals can be mapped directly from the state variables when running onboard. The algorithm is very efficient, but it cannot guarantee the correctness of the mapped control signals. The landing accuracy and optimality are lower than the CVX-based algorithm.

To sum up, CVX and ML algorithms are the two main ways to realize real-time guidance. However, the CVX-based algorithms have the disadvantage of lacking real-time performance. ML-based algorithms have the shortcoming of insufficient landing accuracy and reliability.

It is worth noting that the CVX algorithm is divided into the fixed final-time algorithm and the free final-time algorithm. Since the selection of terminal time significantly impacts the feasibility and optimality of the solution, it is more practical to study the free final-time algorithm. There are two methods in the literature. One is to fix the final time first, adopting LCvx to solve the SOCP, and then regard the final time as a parameter to conduct the one-dimensional golden search [12,23]. Another method treats the final time as a variable and deals with the dynamic constraints by SCvx [24,25]. The relationship between algorithms is shown in Figure 1. The common disadvantage of the two methods is that the SOCP needs to be solved many times, which dramatically reduces the computational efficiency.

**Figure 1.** Classification of the CVX-based algorithms.

Many scholars have optimized the above algorithms. The authors of [26] used quadratic function to approximate the relationship between landing time and landing mass, and reduced the calculation times of the SOCP problem to 4. In reference [27], the grid search method was adopted to calculate the optimal landing time, which reduced the calculation times of the SOCP problem to 6 and was successfully applied in a flight test. The authors of [28] used linear interpolation to directly calculate the optimal landing time through the current state of the rocket, but this method needs to store a large amount of data onboard.

In this paper, the CVX and ML are combined by establishing the DNN predictor to assist convex optimization, thus significantly improving the efficiency of computation (see

the algorithm highlighted in red in Figure 1). The algorithm proposed in this work avoids iteration by introducing the optimal terminal time predictor. Given that the accuracy of the traditional analytical estimation formulas [29,30] is insufficient, inspired by the [19], the DNN is adopted to learn the relationship between state variables and optimal terminal time. The optimal terminal time can be mapped directly from the state variables when running onboard. Therefore, the solution can be obtained by solving the SOCP only once.

The structure of the paper is as follows: Section 1 briefly introduces the background of the powered landing and summarizes the development of guidance algorithms based on CVX and ML. The shortcomings of the current algorithms are illustrated. Section 2 gives the mathematical description of the powered landing. In Section 3, variable substitution and lossless relaxation techniques are used to convexify the primal problem. The effect of terminal time on the feasibility and optimality of the solution is discussed. In Section 4, the DNN optimal predictor is built to predict the optimal terminal time, and the corresponding sample generation algorithm is proposed. Section 5 integrates the algorithms of Sections 3 and 4 and analyzes the optimality and real-time performance by the open-loop simulation. Section 6 proposes a closed-loop guidance algorithm framework based on MPC and analyzes the algorithm's robustness and landing accuracy. Section 7 summarizes the work.

## 2. Problem Statement

Firstly, the reference coordinate system *O-xyz* shown in Figure 2 is established. The origin of the coordinate system is fixed on the landing platform, the *x* axis is aligned to the reverse direction of the local gravity acceleration, and the *y* axis and *z* axis are aligned to the east and north direction of the local ground plane, respectively. Considering the flight time of the powered landing is short, the inertial centrifugal force and Coriolis force caused by the rotation of the reference coordinate system are negligible.

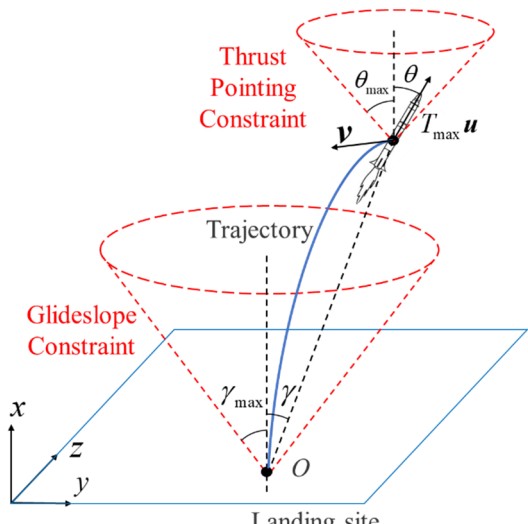

**Figure 2.** Definition of the reference coordinate system for powered landing problem.

It is worth noting that the acceleration provided by the thruster is the main driving force that affects the motion of the rocket, while the aerodynamic force is small relative to the thrust. Therefore, the dynamic model only considers the effect of thrust. In order to evaluate the robustness of the proposed guidance algorithm, an aerodynamic model is added to the closed-loop simulation. The dynamic equation of the 3-DOF motion of the vehicle can be expressed as follows:

$$
\begin{aligned}
\dot{r} &= v \\
\dot{v} &= g + \frac{T}{m} \\
\dot{m} &= -\frac{\|T\|}{I_{sp} g_0}
\end{aligned}
\tag{1}
$$

where $r$, $v$ and $m$ are the position vector, velocity vector and mass of the vehicle, respectively. $I_{sp}$ is the specific impulse of the engine, and $g_0$ is the earth's gravitational constant and $g_0 = 9.8065$ m/s$^2$. $g$ is the gravitational acceleration vector. The initial altitude of the powered landing phase is usually about 5 km, and the variation of the gravitational acceleration is negligible. Therefore, $g$ can be regarded as a constant vector:

$$g = (-g_0, 0, 0)^T \qquad (2)$$

$T$ is the thrust vector and $\|T\|$ is the thrust magnitude. Due to the performance limitation of the rocket engine, $\|T\|$ needs to satisfy the following constraints:

$$T_{min} \leq \|T\| \leq T_{max} \qquad (3)$$

where $T_{min}$ and $T_{max}$ are the lower and upper bounds of the thrust magnitude, respectively. In order to ensure that the vehicle can avoid obstacles and terrain near the landing site, it is also necessary to add the glideslope constraints, as shown in Figure 2:

$$\sqrt{r_y^2(t) + r_z^2(t)} - r_x(t) \tan \gamma_{max} \leq 0 \qquad (4)$$

where $\gamma_{max}$ is the maximum glideslope angle and $\gamma_{max} \in (0, \pi/2)$. $r_x$, $r_y$ and $r_z$ are the three components of the position vector $r$, respectively. Given the constraint, the range of the horizontal motion decreases with altitude, and lateral maneuvers become small as the rocket approaches the landing site. Assuming that the thrust direction is parallel to the longitudinal axis of the rocket, the attitude of the rocket can be limited by constraining the direction of the thrust. The following thrust pointing constraints need to be added, as shown in Figure 2:

$$\sqrt{T_y^2(t) + T_z^2(t)} - T_x(t) \tan \theta_{max} \leq 0 \qquad (5)$$

where $\theta_{max}$ is the maximum thrust pointing angle and $\theta_{max} \in (0, \pi/2)$. $T_x$, $T_y$ and $T_z$ are the three components of the thrust vector, respectively. Given the constraint, the horizontal components of the thrust direction are constrained to a small value, thereby equivalently constraining the attitude of the vehicle.

The initial and terminal constraints are listed as follows:

$$\begin{cases} r(t_0) = r_0 \\ v(t_0) = v_0 \\ m(t_0) = m_0 \\ r(t_f) = 0 \\ v(t_f) = 0 \end{cases} \qquad (6)$$

where $t_0$ and $t_f$ are the initial time and the terminal time of the powered landing, respectively. $r_0$, $v_0$ and $m_0$ are the position, velocity and mass at the initial time, respectively, which can be obtained by the navigation system. Since the landing platform is located at the origin of the coordinate system, the position and velocity vector at the terminal time are both zero vectors.

This work considers the fuel-optimal powered landing problem, so the objective function is selected as the maximum terminal mass:

$$\min J = -m(t_f) \qquad (7)$$

The mathematical description of the fuel-optimal powered landing problem is shown in (1)~(7), which can be summarized in the Figure 3.

**Primal Problem**

Objective function: $\min J = -m(t_f)$

Subject to:

(1) Dynamic constraints

$$\dot{\boldsymbol{r}} = \boldsymbol{v} \quad \dot{\boldsymbol{v}} = \boldsymbol{g} + \frac{\boldsymbol{T}}{m} \quad \dot{m} = -\frac{\|\boldsymbol{T}\|}{I_{sp}g_0}$$

(2) Thrust constraints

$$T_{\min} \le \|\boldsymbol{T}\| \le T_{\max}$$

(3) Thrust pointing constraints

$$\sqrt{T_y^2 + T_z^2} - T_x \tan\theta_{\max} \le 0$$

(4) Glideslope constraints

$$\sqrt{r_y^2 + r_z^2} - r_x \tan\gamma_{\max} \le 0$$

(5) Initial and terminal constraints

$$\boldsymbol{r}(t_0) = \boldsymbol{r}_0 \quad \boldsymbol{v}(t_0) = \boldsymbol{v}_0 \quad m(t_0) = m_0$$
$$\boldsymbol{r}(t_f) = \boldsymbol{0} \quad \boldsymbol{v}(t_f) = \boldsymbol{0}$$

**Figure 3.** Description of the primal problem.

## 3. Lossless Convex Optimization Algorithm

This section first convexifies the primal problem using the variable substitution and lossless relaxation techniques, then discretizes the problem using the equidistant trapezoidal discretization method, and finally converts the problem into an SOCP which can be solved efficiently. Since the selection of terminal time dramatically impacts the optimality and feasibility of the solution, the second part of the section discusses the effect of terminal time on the solution in detail and gives relevant qualitative conclusions.

### 3.1. Convexification and Discretization

From the description of the problem in Section 2, it can be seen that there are some non-convex constraints in the primal problem, which can be divided into two categories. The first is the dynamic non-convex factors caused by the variation of rocket mass with time, which can be convexified by variable substitution. The second is the non-convex factors caused by introducing the lower bound of thrust magnitude, which can be convexified by lossless relaxation. It is proved that after the above convex steps, the optimal solution of the problem is the same as primal problem [11,12].

First, define the logarithm of mass as a new variable to replace rocket mass:

$$z(t) = \ln[m(t)] \tag{8}$$

Similarly, define new control variables:

$$u(t) = \frac{T(t)}{m(t)} \quad \sigma(t) = \frac{\|T(t)\|}{m(t)} \tag{9}$$

where $u(t)$ and $\sigma(t)$ are the defined control variables, which indicate the thrust acceleration components and magnitude, respectively. Then the mass flow constraint is rewritten as follows:

$$\dot{z}(t) = \frac{\dot{m}(t)}{m(t)} = -\frac{\|T(t)\|}{I_{sp}g_0 m(t)} = -\frac{\sigma(t)}{I_{sp}g_0} \tag{10}$$

New dynamic constraints and mass flow constraints are derived:

$$\begin{aligned} \dot{r} &= v \\ \dot{v} &= g + u \\ \dot{z} &= -\frac{\sigma}{I_{sp}g_0} \end{aligned} \tag{11}$$

The upper and lower bound constraints of the thrust magnitude are transformed into the following formula:

$$T_{\min} \le \|T(t)\| \le T_{\max} \Rightarrow \frac{T_{\min}}{m(t)} \le \|\frac{T(t)}{m(t)}\| \le \frac{T_{\max}}{m(t)} \Rightarrow T_{\min}e^{-z(t)} \le \sigma(t) \le T_{\max}e^{-z(t)} \tag{12}$$

The Taylor expansion is used to simplify the expression further, and only the first-order term is retained to obtain the approximate constraint as:

$$T_{\min}e^{-z_0(t)}[1-(z(t)-z_0(t))] \leq \sigma(t) \leq T_{\max}e^{-z_0(t)}[1-(z(t)-z_0(t))] \tag{13}$$

where $z_0(t)$ is the lower bound of the $z(t)$. The specific expression of $z_0(t)$ is as follow:

$$z_0(t) = \ln\left(m_0 - \frac{T_{\max}}{I_{sp}g_0}t\right) \tag{14}$$

It is worth noting that only retaining the first-order term will introduce the linearization error. However, according to the numerical simulation in Section 5, it can be verified that the error is small. So far, the upper and lower bound constraints of the thrust magnitude have been transformed into linear inequality constraints. Because there is a non-convex constraint relationship between the thrust acceleration component and magnitude, lossless relaxation is required:

$$\|u(t)\| = \sigma(t) \Rightarrow \|u(t)\| \leq \sigma(t) \tag{15}$$

Formula (15) is called the lossless relaxation technique, and it is a key step in convexification. Although (15) makes the constraint convex, it increases the feasible domain of the primal problem. Refs. [11,12] and a large number of numerical simulations show that although the feasible domain of the problem has changed after relaxation, the optimal solutions before and after relaxation are equivalent. The mathematical description of the fuel-optimal powered landing problem can be summarized in the Figure 4.

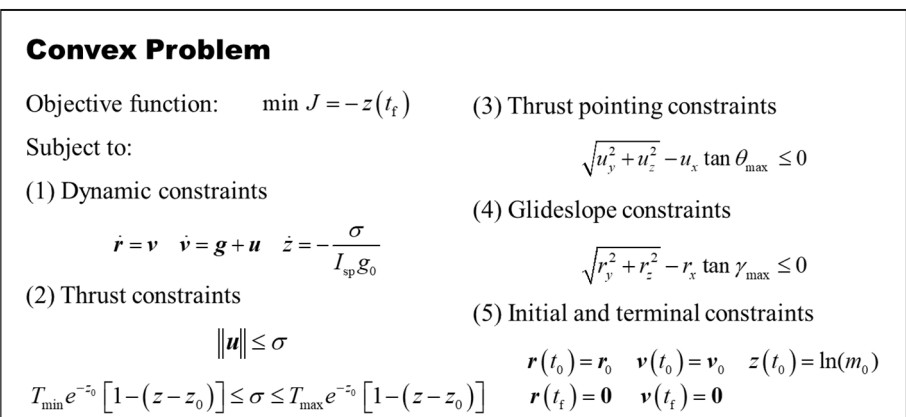

**Figure 4.** Description of the convex problem.

The convex problem needs to be further discretized to transform the infinite-dimensional optimization problem into a finite-dimensional problem. The computational efficiency and accuracy of different discrete methods are various, discussed in detail in reference [15]. In order to simplify the derivation and highlight the essence of the problem, the equidistant trapezoidal discretization method is adopted.

The given time interval $[0, t_f]$ is uniformly discretized into $N$ points, and the length of each discrete segment is $\Delta t = t_f/(N-1)$. Define the vector to be optimized as:

$$x[k] = \left(r[k]^T, v[k]^T, z[k], u[k]^T, \sigma[k]\right)^T \qquad k = 1, \ldots, N \tag{16}$$

Then, the dynamic equation constraints in discrete form can be written according to (11):

$$
\begin{aligned}
r[k] - r[k+1] + \tfrac{\Delta t}{2} v[k] + \tfrac{\Delta t}{2} v[k+1] &= 0 \\
v[k] - v[k+1] + \tfrac{\Delta t}{2} u[k] + \tfrac{\Delta t}{2} u[k+1] &= -g\Delta t \qquad k = 1,\dots,N-1 \\
z[k] - z[k+1] - \tfrac{\Delta t}{2 I_{sp} g_0} \sigma[k] - \tfrac{\Delta t}{2 I_{sp} g_0} \sigma[k+1] &= 0
\end{aligned}
\tag{17}
$$

Since the objective function of the discretized problem is a linear function, the equality constraints are affine, and the inequality constraints are a combination of affine constraints and second-order cone constraints. The original continuous optimization problem is completely transformed into a finite-dimensional SOCP. The results can be summarized in the Figure 5.

**Discrete Convex Problem**

Objective function: $\quad \min J = -z[N]$

Variables: $x[k] = \left( r[k]^T, v[k]^T, z[k], u[k]^T, \sigma[k] \right)^T$

Subject to:

(1) Affine equality constraints

$$ r[k] - r[k+1] + \frac{\Delta t}{2} v[k] + \frac{\Delta t}{2} v[k+1] = 0 $$

$$ v[k] - v[k+1] + \frac{\Delta t}{2} u[k] + \frac{\Delta t}{2} u[k+1] = -g\Delta t $$

$$ z[k] - z[k+1] - \frac{\Delta t}{2 I_{sp} g_0} \sigma[k] - \frac{\Delta t}{2 I_{sp} g_0} \sigma[k+1] = 0 $$

$$ r[1] = r_0 \quad v[1] = v_0 \quad z[1] = \ln(m_0) $$

$$ r[N] = 0 \quad v[N] = 0 $$

(2) Affine inequality constraints

$$ T_{\min} e^{-z_0[k]} \left[ 1 - \left( z[k] - z_0[k] \right) \right] \le \sigma[k] $$

$$ \sigma[k] \le T_{\max} e^{-z_0[k]} \left[ 1 - \left( z[k] - z_0[k] \right) \right] $$

(3) Second order cone constraints

$$ \sqrt{u_x^2[k] + u_y^2[k] + u_z^2[k]} \le \sigma[k] $$

$$ \sqrt{u_y^2[k] + u_z^2[k]} \le u_x[k] \tan\theta_{\max} $$

$$ \sqrt{r_y^2[k] + r_z^2[k]} \le r_x[k] \tan\gamma_{\max} $$

**Figure 5.** Description of the discrete convex problem.

The problem can be solved efficiently using a mature SOCP solver. All numerical simulations in the paper are performed on an Intel Core i7-10875H PC with 2.3 GHz and 64 GB memory using the ECOS solver [31]. The algorithm used by the solver is a standard primitive-dual, prediction-correction interior-point algorithm, using Nesterov-Todd (NT) scaling and self-dual embedding technology. ECOS is a small-footprint, high-performance SOCP solver whose accuracy is numerically reliable and beyond that typically required by embedded applications. It is competitive in solving small and medium-sized convex optimization problems.

*3.2. Discussion about Optimal Terminal Time*

Section 3.1 uses the LCvx to convert the problem into an SOCP and efficiently solve it. However, the algorithm requires a given terminal time in advance, which is also the main shortcoming of the algorithm in Section 3.1. Nevertheless, few scholars have studied the impact of terminal time on the solution in detail. This section will analyze the relationship between the terminal mass and the terminal time through the simulation.

First, set the rocket parameters and environment parameters, as shown in Table 1.

**Table 1.** Rocket and environmental parameters.

| Parameter | Amount | Parameter | Amount |
|---|---|---|---|
| $T_{min}$ | 845.2 kN | $I_{sp}$ | 282 s |
| $T_{max}$ | 169.0 kN | $\gamma_{max}$ | 80° |
| $g_0$ | 9.80665 m/s$^2$ | $\theta_{max}$ | 30° |

Then, set the typical initial state variables, as shown in Table 2.

**Table 2.** Setting of initial state variables.

| Parameter | Amount | Parameter | Amount |
|---|---|---|---|
| $r_{x0}$ | $5 \times 10^3$ m | $v_{x0}$ | $-150$ m/s |
| $r_{y0}$ | $5 \times 10^2$ m | $v_{y0}$ | $-30$ m/s |
| $r_{z0}$ | $5 \times 10^2$ m | $v_{z0}$ | 30 m/s |
| $m_0$ | $3.8 \times 10^4$ kg | | |

The relationship between the terminal mass and the terminal time is obtained using the ECOS solver, as shown in Figure 6.

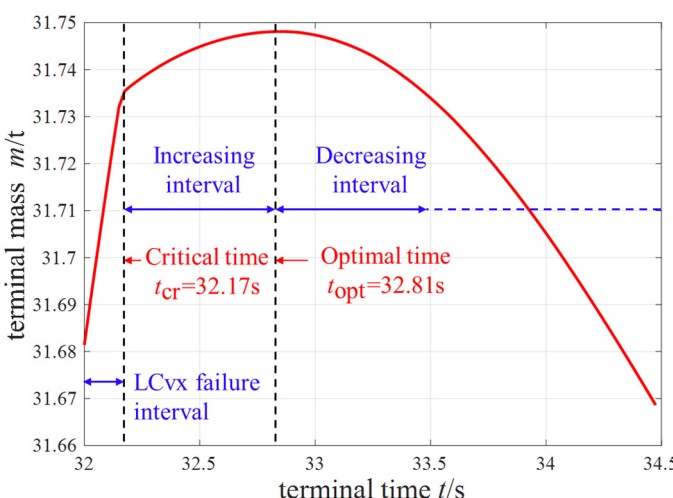

**Figure 6.** The relationship between terminal mass and terminal time.

The relationship can be divided into three intervals:

1. LCvx failure interval ($t_f \in [0, t_{cr}]$): Although it is possible for the numerical solver to obtain the optimal solution, the convex relaxation is no longer valid, which means the solutions do not satisfy $\|u(t)\| = \sigma(t)$. Therefore, the solution is infeasible;
2. Increasing interval ($t_f \in (t_{cr}, t_{opt}]$): The algorithm can successfully obtain the optimal solution, and the terminal mass increases monotonously with the terminal time. Numerical simulations show that this interval is usually narrow. In Figure 6, the interval length is only 0.64 s. When the initial state variables are set to other values, the interval length is generally less than 2 s;
3. Decreasing interval ($t_f$ is slightly larger than $t_{opt}$): The algorithm can also successfully obtain the optimal solution, and the terminal mass decreases monotonously with the terminal time. The rocket will increase the landing time by consuming more fuel in this interval. If $t_f$ is very large, the altitude might increase during the powered landing.

Combining the above three situations, the following conclusions can be drawn. The feasibility and optimality of the algorithm in Section 3.1 are closely related with $t_f$. If $t_f$ is too small, a feasible trajectory cannot be obtained. If the value is too large, it will consume more fuel, and even lead to increasing the altitude. In addition, it can be found

that the $t_{\text{opt}}$ and $t_{\text{cr}}$ are very close, so choosing an inappropriate $t_{\text{f}}$ can easily lead to an infeasible solution.

Many scholars used the one-dimensional search and SCvx method to solve the $t_{\text{opt}}$, which have achieved good results. However the two methods both need to solve the SOCP many times, and the number of iterations is related to the initial selection of $t_{\text{f}}$. Since the number of iterations cannot be predicted in advance when running onboard, and the iteration significantly increases the computation cost, the above iteration-based method limits the algorithm's efficiency in Section 3.1.

## 4. DNN Optimal Predictor

Section 3.2 illustrates the shortcomings of traditional iteration-based algorithms. It is necessary to propose a new method to avoid iteration.

Feedforward neural network is the earliest artificial neural network. It has a robust fitting ability and can be used to fit the nonlinear relationship between all kinds of data. According to the Universal Approximation Theorem, for a feedforward neural network consisting of a linear output layer and at least one hidden layer using nonlinear activation functions, as long as there are enough neurons in the hidden layer, then it can approximate a bounded closed set function defined in real space with arbitrary accuracy.

There is a complex implicit relationship between the initial state and the optimal terminal time for the fuel-optimal powered landing problem, so it is very suitable to use a feedforward neural network to fit the relationship between them. If the number of hidden layers increases, the network can fit more complex nonlinear relationships with fewer neurons. Therefore, the deep feedforward neural network is chosen to build the optimal predictor, referred to as DNN.

This section aims to replace the traditional iteration-based method by introducing the DNN $t_{\text{opt}}$ predictor. The predictor can accurately and efficiently map the optimal landing time according to the current state variables, which significantly improves the computational efficiency of the LCvx algorithm with free terminal time. The specific algorithm denoted as DNN-LCvx is shown in Figure 7.

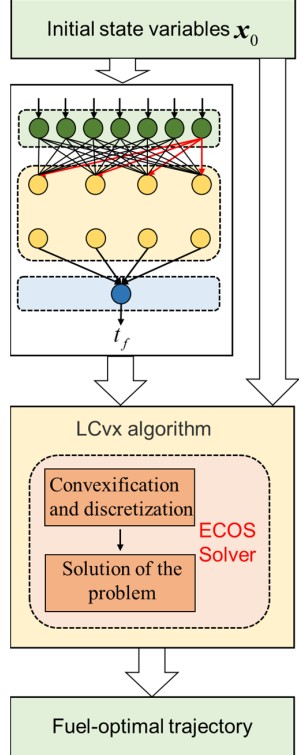

**Figure 7.** Flow chart of the DNN-LCvx algorithm.

### 4.1. Sample Generation

The key to constructing the DNN optimal predictor is to generate a large number of good training samples. In this paper, the SCvx algorithm proposed in [25] is used to generate the sample trajectory efficiently. Then, the state variables and the corresponding optimal terminal time are obtained directly using linear interpolation. The sample trajectory's initial state variables are generated as follows:

$$
\begin{aligned}
&r_{x,i}= 5000 + 1000\varepsilon \ (\mathrm{m}) &&v_{x,i}= -100 + 100\varepsilon \ (\mathrm{m/s}) \\
&r_{y,i}= 1000\varepsilon \ (\mathrm{m}) &&v_{y,i}= 50\varepsilon \ (\mathrm{m/s}) \\
&r_{z,i}= 1000\varepsilon \ (\mathrm{m}) &&v_{z,i}= 50\varepsilon \ (\mathrm{m/s}) \\
&m_{i}= 38 + 2\varepsilon \ (\mathrm{t}) &&\varepsilon \in [-1,1]
\end{aligned}
\tag{18}
$$

where $\varepsilon$ is a uniformly distributed random number with the range of $[-1, 1]$. First, the corresponding fuel-optimal trajectory is solved by the SCvx algorithm for each pair of initial state variables. Then, 60 state variables and optimal terminal time pairs are uniformly selected as samples on each sample trajectory. Finally, the generated samples are stored in the training set. The flow chart of sample generation is shown in Figure 8.

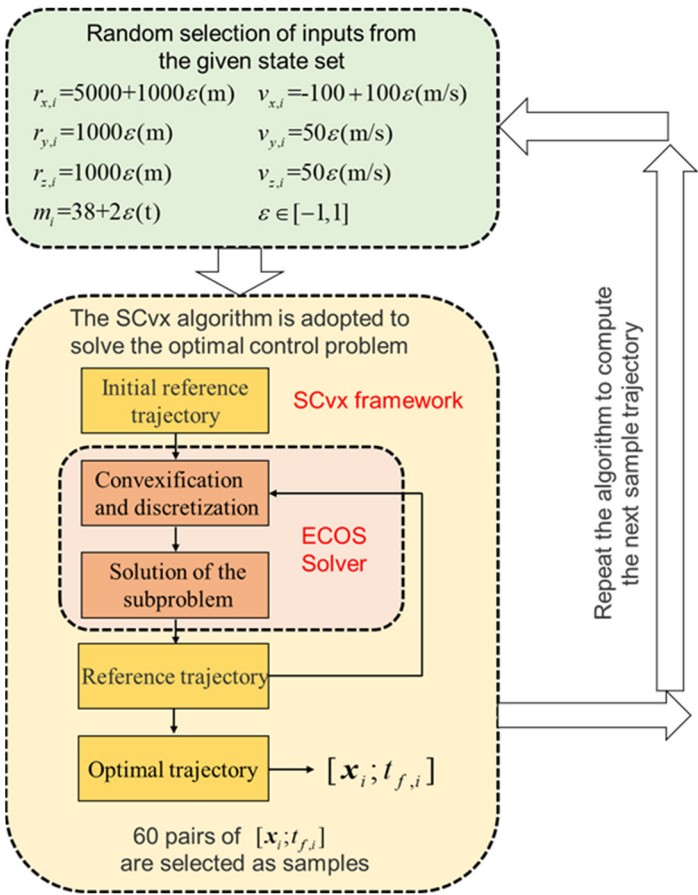

**Figure 8.** Flow chart of the sample generation.

A large number of sample trajectories can be generated efficiently using the above algorithms. A total of 20,000 sample trajectories are randomly generated, and the sample trajectories are shown in Figure 9. Because 60 pairs of samples are uniformly distributed on each sample trajectory, the total number of the samples is 1.2 million, which basically covers the whole feasible region.

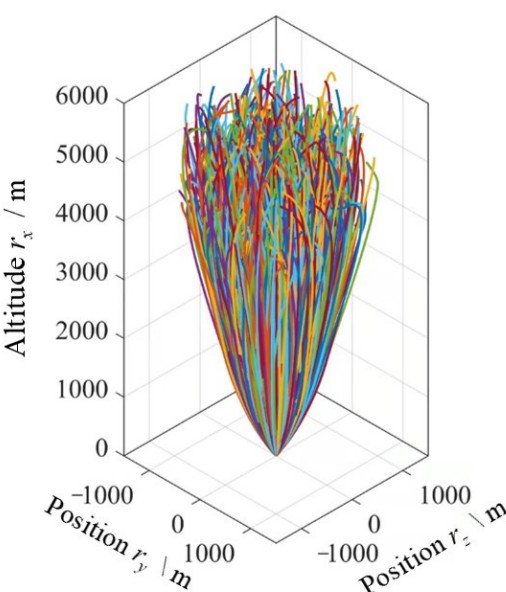

**Figure 9.** Distribution of the sample trajectories.

The distribution of the terminal time of all sample trajectories is drawn, as shown in Figure 10. It can be found that there is a great variant between the optimal terminal time obtained under different initial conditions. Among the 20,000 sample trajectories, the minimum optimal terminal time is 25.1219 s, and the maximum is 57.8355 s, and the sample data is reasonable.

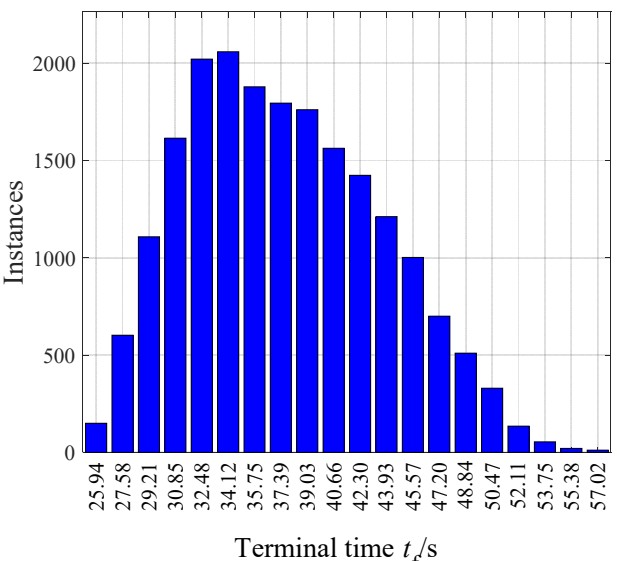

**Figure 10.** Statistics of the optimal terminal time of sample trajectories.

*4.2. Design and Training of the DNN*

The DNN structure used in this section is shown in Figure 11.

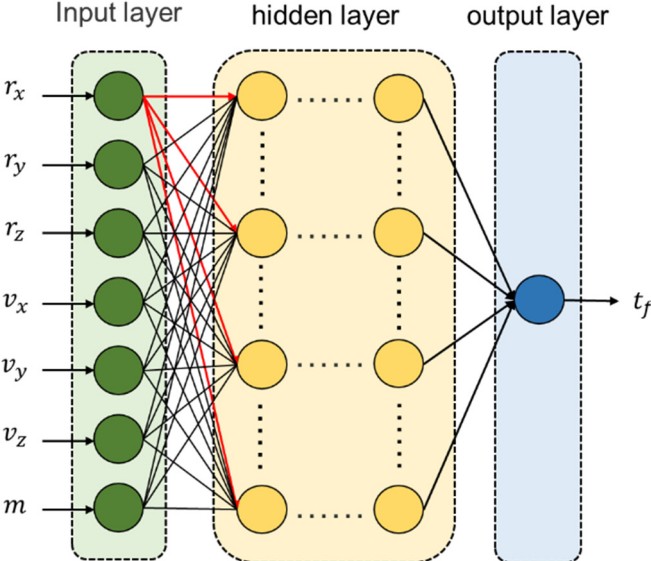

**Figure 11.** Structure of the DNN.

The input layer of the DNN contains 7 neuron units, which represent the current position components, velocity components, and mass of the vehicle, respectively. The output layer contains 1 unit representing the estimation of $t_{opt}$. Each unit in the network structure is essentially a nonlinear function that describes the mapping relationship between input and output. The number of hidden layers, the number of units in each layer, and the activation function are all hyperparameters of the DNN, which need to be determined by optimizing the network structure. Define a loss function in the form of mean squared error (MSE) as follows:

$$\sigma_{\text{MSE}} = \frac{\sum\limits_{i=1}^{n} \left( t_{i,\text{opt}} - \hat{t}_{i,\text{opt}} \right)^2}{n} \tag{19}$$

where $n$ is the number of samples input for each training, $t_{i,\text{opt}}$ and $\hat{t}_{i,\text{opt}}$ are the true values and estimated values of the optimal terminal time, respectively. Similarly, in order to measure the error of the network, an error function in the form of mean absolute error (MAE) is defined:

$$\sigma_{\text{MAE}} = \frac{\sum\limits_{i=1}^{n} \left| t_{i,\text{opt}} - \hat{t}_{i,\text{opt}} \right|}{n} \tag{20}$$

The Bayesian regularized backpropagation algorithm is selected to train the DNN, and the method updates the weights and biases according to the Levenberg–Marquardt optimization algorithm. Bayesian regularization minimizes the linear combination of squared errors and weights and modifies the linear combination to give a network that generalizes well after training. The neural network hyperparameters are shown in Table 3:

**Table 3.** Setting of the DNN's hyperparameters.

| Parameter | Amount | Parameter | Amount |
|---|---|---|---|
| Number of hidden layers | 2 | Training ratio | 85% |
| Number of hidden layer units | 30 | Test ratio | 15% |
| Activation function | Sigmoid | Maximum epochs | 1000 |

The variation of the training set and the test set's MSE with the training epochs is shown in Figure 12:

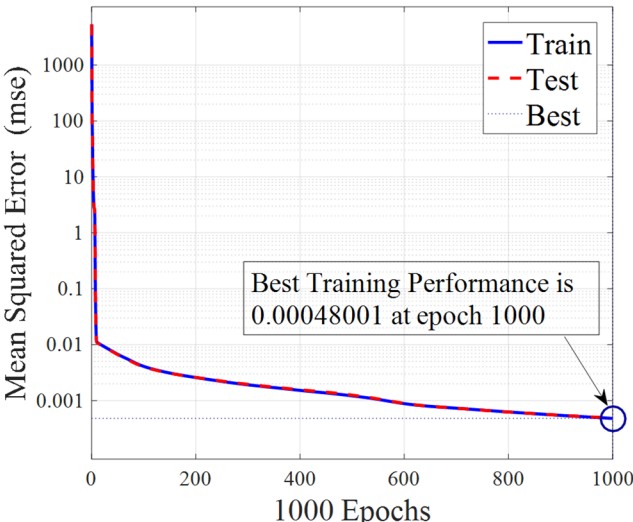

**Figure 12.** Visualization of the training process.

It can be found that the MSE decreases monotonically with the epoch. When the epoch reaches 1000, the MSE of the training set reaches the minimum, and $\sigma_{\mathrm{MSE}} = 4.8 \times 10^{-4}\ \mathrm{s}^2$. The MSE of the training set and the test set are highly coincident, indicating that the generalization ability of the net is good, and there is no overfitting in the training results.

After training, the distribution of the network error is plotted, as shown in Figure 13.

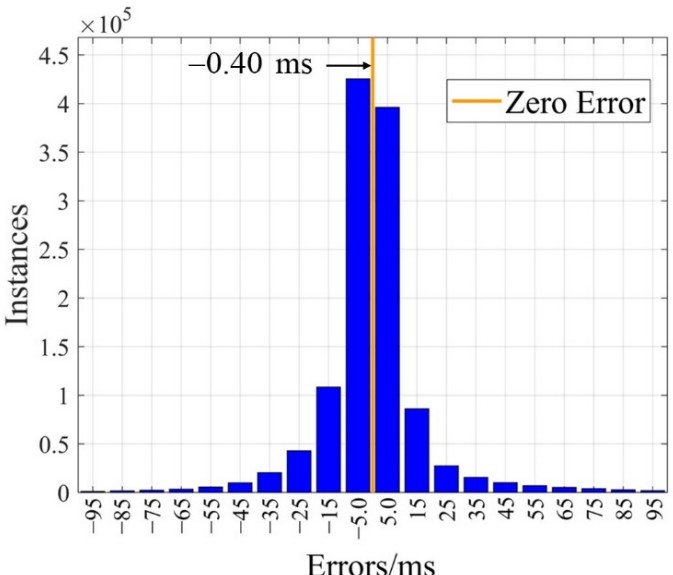

**Figure 13.** Statistics of the DNN errors.

It can be found that the DNN constructed in this paper can accurately predict the optimal terminal time according to the current state. The MAE is 11.6 ms, and the prediction errors of most samples (99.11%) do not exceed 0.1 s. The specific results are shown in Table 4.

**Table 4.** Statistics of training results.

| Parameter | Amount | Parameter | Amount |
| --- | --- | --- | --- |
| Mean square error (MSE) | $4.8074 \times 10^{-4}\ \mathrm{s}^2$ | Mean absolute error (MAE) | 11.6 ms |
| MSE of the training set | $4.8001 \times 10^{-4}\ \mathrm{s}^2$ | Maximum error | 0.3921 s |
| MSE of the test set | $4.8488 \times 10^{-4}\ \mathrm{s}^2$ | Minimum error | −0.4530 s |

## 5. Open-Loop Simulation

In this section, the DNN-LCvx algorithm (see Figure 7) is verified by open-loop numerical simulations. The primary purpose is to verify the algorithm's optimality and real-time performance and discuss the possibility of its application to onboard guidance.

### 5.1. Optimality Performance

The SCvx algorithm proposed in [25] is regarded as a standard algorithm, and the errors between the DNN-LCvx and the SCvx are analyzed. If the control profiles computed by the two algorithms are similar, and the terminal landing mass and terminal time are close, it can be considered that the DNN-LCvx proposed in this paper has good optimality performance.

The initial state variables are shown in Table 2. The two algorithms are discretized with uniform time interval, and $N = 100$. Figures 14 and 15 show the control profiles of the two algorithms.

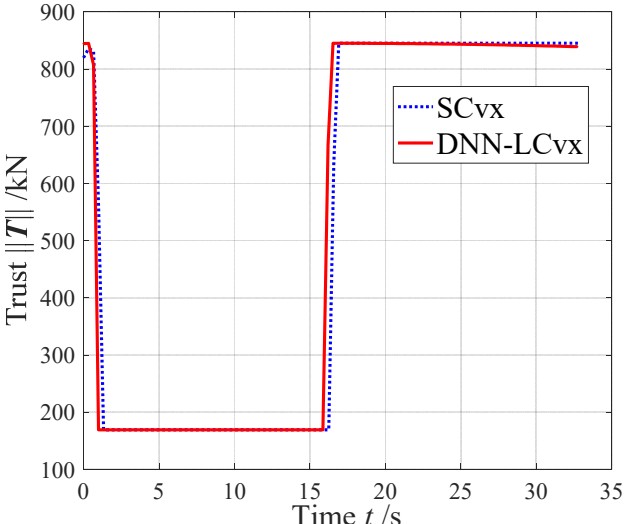

**Figure 14.** Comparison of the thrust amplitude profiles between the SCvx and DNN-LCvx.

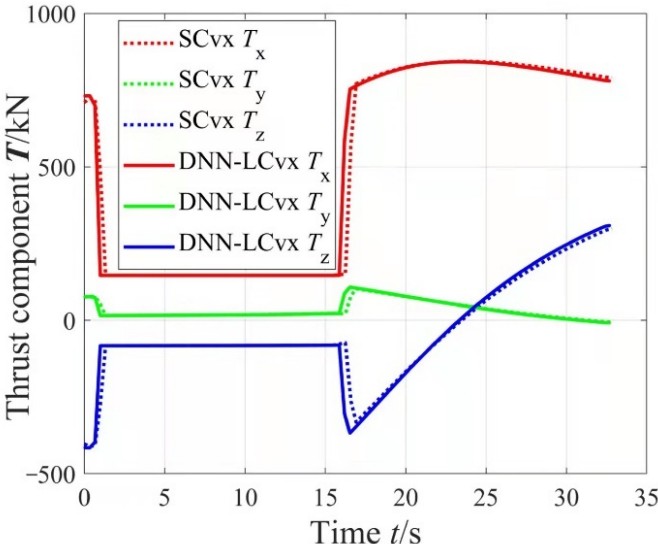

**Figure 15.** Comparison of the thrust component profiles between the SCvx and DNN-LCvx.

It can be found that the control profiles computed by the two algorithms are similar. The optimal terminal time solved by SCvx is 32.815 s and the terminal mass is 31,760.5 kg. The optimal terminal time solved by the DNN-LCvx algorithm is 32.719 s and the terminal mass is 31,755.7 kg. The terminal time error is −0.096 s and the terminal mass error is −4.8 kg. The errors are at a small level. Therefore, the optimality of the DNN-LCvx algorithm can be verified.

Next, the impact of the discrete number on optimality is discussed. Generally, the larger the discrete number is, the closer the result is to the optimal solution, but the computation cost will increase significantly. The variation curve of terminal mass with the discrete number is plotted in Figure 16.

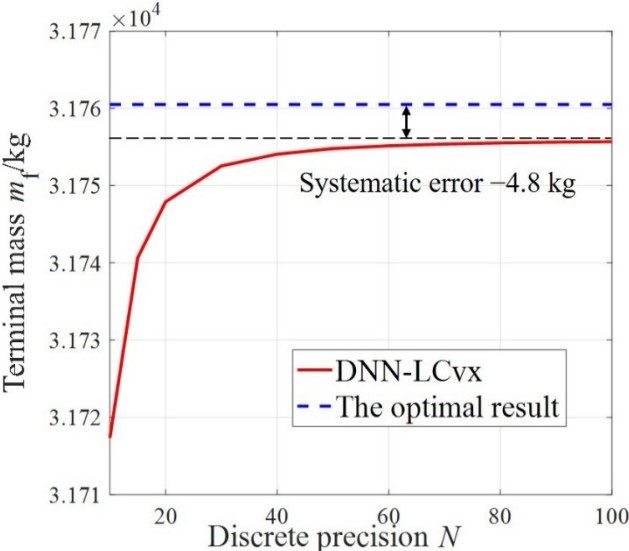

**Figure 16.** Influence of the discrete precision on terminal mass.

It can be seen from Figure 16 that there is a systematic error in the DNN-LCvx algorithm, which is mainly caused by the linearization of (13) and the error of the network. It is worth noting that when the discrete number is greater than 30, the terminal mass changes very smoothly with $N$. Therefore, $N > 30$ can ensure the optimality of the algorithm.

### 5.2. Real-Time Performance

This section evaluates the real-time performance of the DNN-LCvx algorithm by comparing the elapsed time of the proposed DNN-LCvx algorithm and the standard SCvx algorithm. The initial state variables are shown in Table 2. The discrete number is $N = 30$, and the general ECOS solver is used to solve the SOCP. The elapsed time and other information obtained from the test are shown in Table 5.

**Table 5.** Comparison of real-time performance between the SCvx and DNN-LCvx.

| Item | DNN-LCvx | SCvx |
| --- | --- | --- |
| Elapsed time (ms) | 6.5 | 78.0 |
| Iterations | 1 | 5 |
| Elapsed time per Iteration (ms) | 6.5 | 15.6 |

It can be found that the DNN-LCvx has good real-time performance. In this example, the DNN-LCvx algorithm is 12.0 times faster than the SCvx algorithm, which is mainly caused by two factors:

1.   SCvx adopts the idea of solving problem in sequence, so it needs to solve sub-SOCPs many times, which significantly increases the elapsed time, while DNN-LCvx can be completed only once.

2.  In order to prevent the failure of the linearization and the artificial infeasibility, the SCvx algorithm needs to introduce a large number of trust region constraints and virtual control variables, thus increasing the complexity of the sub-SOCP.

In summary, the DNN-LCvx algorithm proposed in this paper has excellent real-time performance compared with traditional algorithms. Next, we discuss the impact of the discrete number on real-time performance. The variation of the elapsed time with the discrete number is plotted, as shown in Figure 17. If the discrete number is the same, the computation speed of DNN-LCvx is about an order of magnitude higher than SCvx.

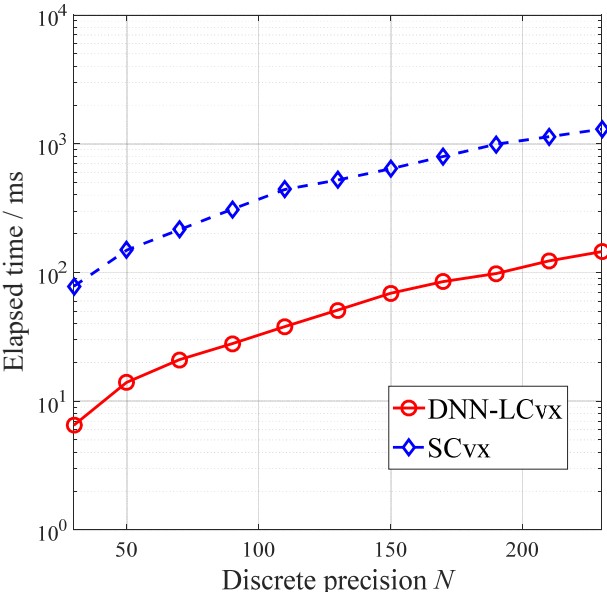

**Figure 17.** Impact of the discrete precision on elapsed time.

## 6. Closed-Loop Simulation

The simulations in Section 5 are only open-loop simulations for the ideal dynamic model. However, there are complex environmental disturbances and thrust execution errors in real flight. Therefore, it is necessary to introduce closed-loop control to achieve a high-precision landing. In this section, a closed-loop guidance algorithm based on MPC is proposed. The proposed algorithm's accuracy and robustness are verified by adding aerodynamic drag, thrust execution error, and initial state errors to the model.

In the implementation of the MPC-based closed-loop guidance algorithm, the navigation system first obtains the state variables, including the flight velocity, position, and the remaining mass. The obtained data is then transmitted to the DNN optimal predictor, which estimates the optimal landing time. Then, the state variables and the optimal landing time are transmitted to LCvx to obtain the optimal control profile. Finally, the average value of the control variables in the first guidance period is transmitted to the dynamic model, and the state variables are updated. This process will repeat after one guidance period until the rocket safely reaches the landing site. The algorithm flow chart is shown in Figure 18.

### 6.1. Adaptability to Environment and Control System

In order to guarantee convergence and computational efficiency, the DNN-LCvx algorithm does not consider the atmospheric drag. This section will verify the robustness of the proposed algorithm by adding an atmospheric drag term to the dynamics model. Moreover, the effect of the control system execution error on the simulation results is also considered. The thrust in the guidance period is assumed to be a constant value that cannot be continuously changed. The constant value is the interpolation average of the optimal thrust sequence in the guidance period. In the closed-loop simulation, the

increased disturbances will cause the actual state variables to diverge from the planned trajectory. Therefore, the guidance module needs to update the guidance law in real-time, as shown in Figure 18.

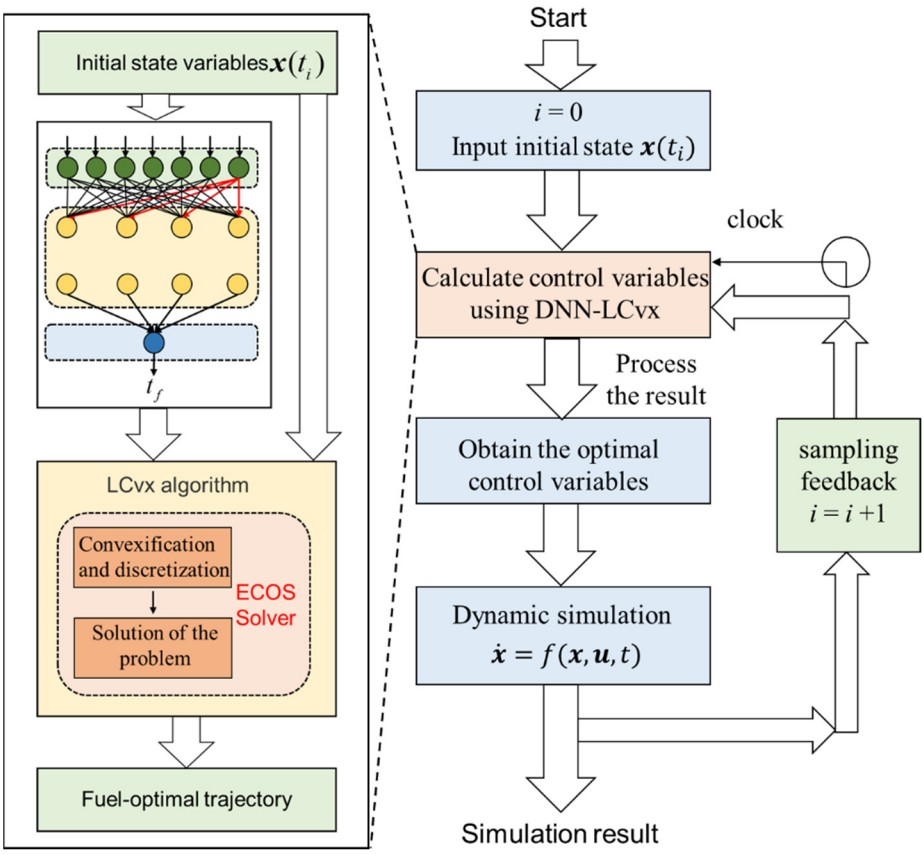

**Figure 18.** Flow chart of the MPC-based closed-loop simulation.

The dynamic equations of the system are:

$$\dot{r} = v$$
$$\dot{v} = g + \frac{\kappa T}{m} - \frac{1}{2m}\rho C_d S_{ref}\|v\|v$$
$$\dot{m} = -\frac{\kappa\|T\|}{I_{sp}g_0}$$

(21)

where $\rho$ is the density of the atmosphere, and the value is 1.225 kg/m³; $C_d$ is the drag coefficient; $S_{ref}$ is the reference area of the rocket, and the value is 10.51 m². $T$ is the control variables computed by DNN-LCvx, and $\kappa$ is the ratio of the actual control variables to the expected control variables, which indicates the error of the control system. The remaining parameters are the same as (1), and the initial conditions of the simulation are shown in Table 2. The simulation results of the landing error of different $C_d$ and $\kappa$ can be obtained, as shown in Tables 6 and 7, where the guidance period is 200 ms and the discrete number is 30.

**Table 6.** Statistics of landing error with $\kappa$ = 1.0.

| $C_d$ | 0.5 | 1.0 | 1.5 | 2.0 |
|---|---|---|---|---|
| position error (m) | 0.0480 | 0.0116 | 0.0133 | 0.0189 |
| velocity error (m/s) | 0.1701 | 0.0012 | 0.0027 | 0.0147 |

**Table 7.** Statistics of landing error with $C_d = 1.0$.

| $\kappa$ | 0.95 | 0.98 | 1.02 | 1.05 |
|---|---|---|---|---|
| position error (m) | 0.8299 | 0.3573 | 0.0025 | 0.0182 |
| velocity error (m/s) | 1.7474 | 0.5079 | 0.0201 | 0.0755 |

It can be seen from above results that when $\kappa$ is fixed and $C_d$ is free, the rocket can achieve high-precision landing, and the landing accuracy is not highly correlated with the $C_d$. When $C_d$ is fixed and $\kappa$ is free, the rocket can also achieve high-precision landing. However, the simulation also indicated that the landing error would increase significantly for $\kappa < 1$. For example, the landing speed error reached 1.7474 m/s if $\kappa = 0.95$.

The following strategy can be used to solve the problem of low landing accuracy for $\kappa < 1$: In the trajectory planning stage, set $T_{max}$ slightly smaller than the nominal maximum thrust $T_{n,max}$ in advance. When the rocket's maximum thrust is smaller than $T_{n,max}$, there is still enough margin to guarantee $\kappa \geq 1$.

The thrust profile is plotted with $C_d = 1.0$ and $\kappa = 1.02$, as shown in Figure 19.

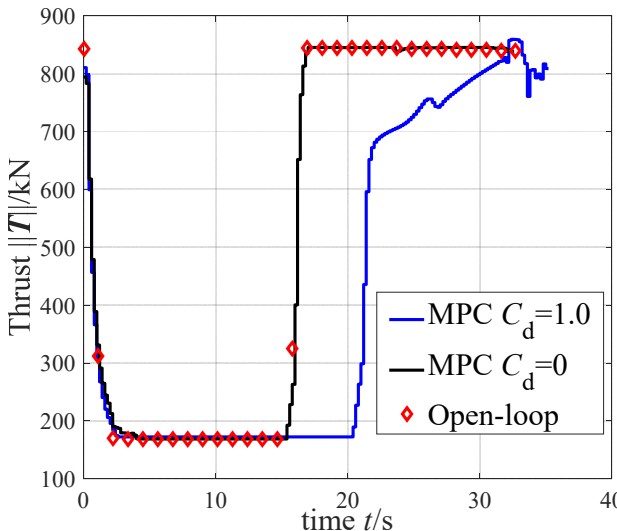

**Figure 19.** Comparison of the closed-loop thrust magnitude profiles between different disturbances.

The red marks in the figure indicate the open-loop control sequence running the DNN-LCvx algorithm at the initial state. The black curve is the closed-loop simulation result without any disturbance, and it can be seen that the closed-loop simulation results are highly consistent with the open-loop results. The blue curve is the control sequence obtained by closed-loop simulation considering aerodynamic drag and thrust execution error. It can be seen that the second switch time of the thrust is delayed, and the total flight time of the rocket also increases accordingly. The position error of the landing is 0.0025 m, and the velocity error is 0.0201 m/s. The algorithm can achieve a high-precision landing.

It is worth noting that the final landing mass obtained by the open-loop trajectory planning is 31.752 t. When the aerodynamic drag is added, the final landing mass obtained by the closed-loop simulation is 32.616 t. The fuel consumption is reduced by nearly 1 t because aerodynamic drag has the positive effect of slowing the rocket's velocity, reducing the fuel consumption required for the landing.

### 6.2. Adaptability to Initial States

This section mainly analyzes the robustness of the closed-loop algorithm under different initial states. According to the analysis in Section 4.1, if the initial state satisfies (18), the optimal landing time obtained by the DNN optimal predictor is reliable. Therefore, this section randomly selects 150 initial state combinations according to (18), and then performs a Monte Carlo simulation to analyze the landing accuracy.

Selecting $C_d = 1.0$ and $\kappa = 1.02$, the scatter plot of position error and velocity error in the horizontal direction can be obtained, as shown in Figure 20.

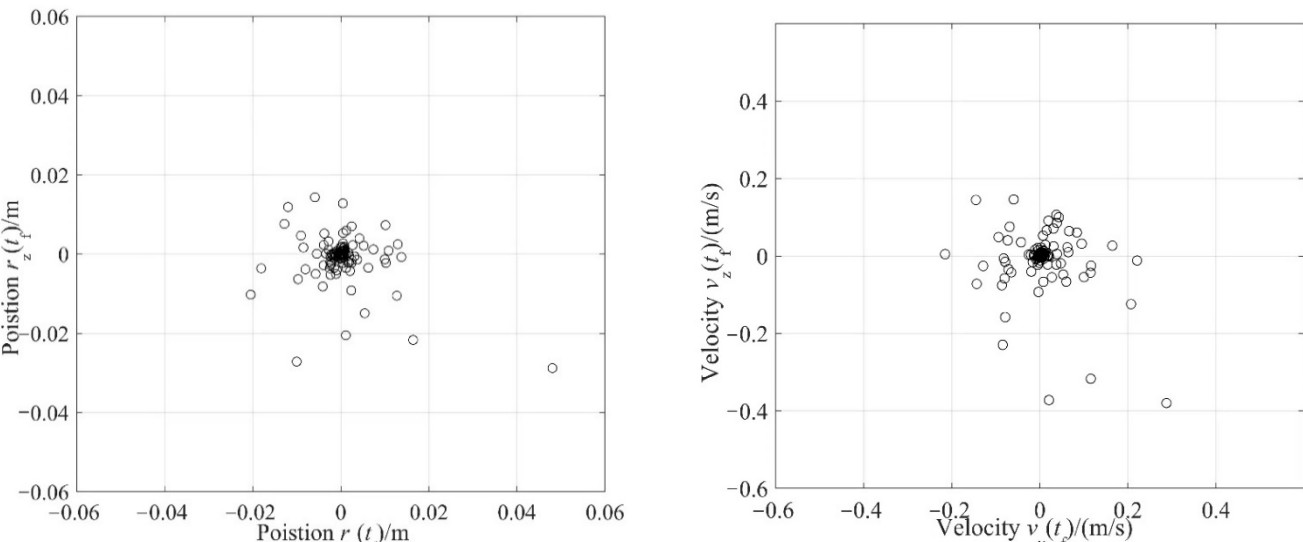

**Figure 20.** Errors of the horizontal position and velocity.

Figure 20 shows that the rocket can achieve precision landing for all initial states. The position error of the landing in the horizontal direction is within 0.06 m, and the velocity error of the landing in the horizontal direction is within 0.4 m/s. It meets the requirement of safe landing. In addition, the excessive vertical speed at the terminal time will cause damage to the rocket structure, so it is also necessary to analyze the vertical speed of landing, as shown in Figure 21.

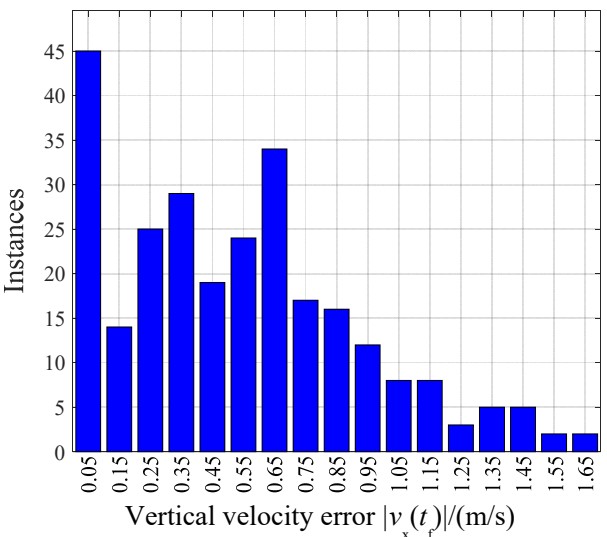

**Figure 21.** Errors of the vertical velocity.

The simulation results show that the vertical velocity of landing is less than 1.7 m/s, and most of the numerical examples are less than 1 m/s. The rocket trajectories under the random initial state are shown in Figure 22. The position components' profiles under the random initial state are shown in Figure 23.

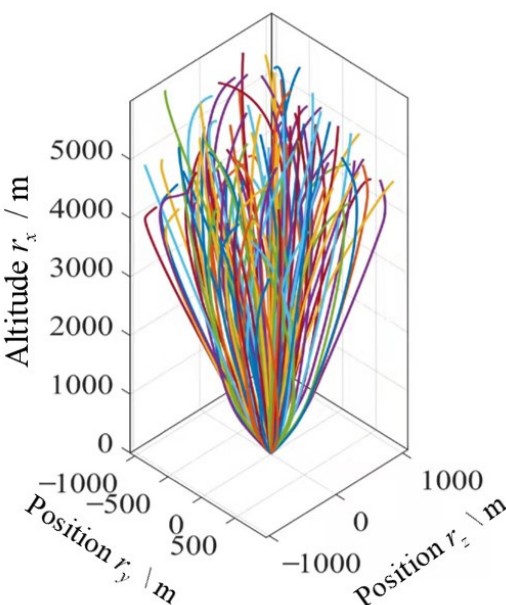

**Figure 22.** Closed-loop simulation trajectory with random initial state variables.

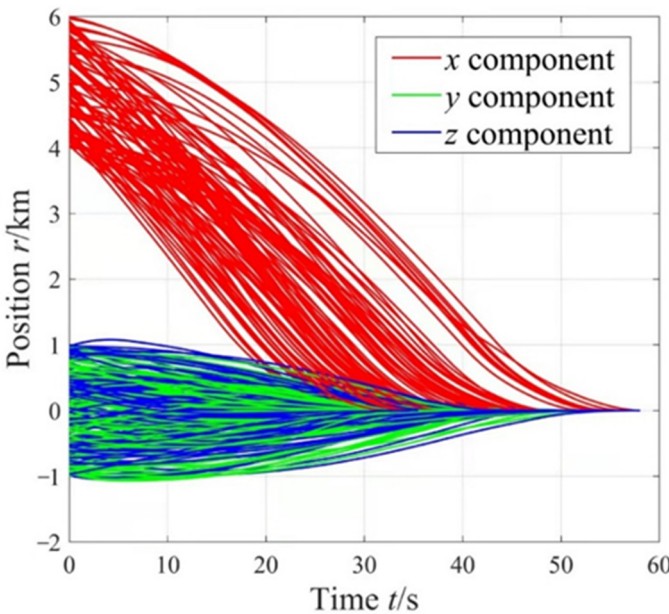

**Figure 23.** Closed-loop position profiles with random initial state variables.

The simulation results show that the rocket can accurately land under different initial conditions. Figure 23 shows that the landing time varies greatly under different initial values, with a maximum variety of nearly 40 s. The algorithm proposed in this paper can effectively and reliably solve the optimal landing time.

## 7. Conclusions

This work mainly studies an efficient guidance algorithm for fuel-optimal powered landing. The LCvx algorithm is combined with the DNN predictor to construct the DNN-LCvx algorithm, accurately predicting the optimal landing time and efficiently solving the optimal trajectory with various constraints. Open-loop numerical simulation shows that:

1.  The computation accuracy of the proposed algorithm is high, and the error of terminal mass is several kilograms, which can ensure the optimality of the results.

2. The computational efficiency of the proposed algorithm is high, and it is about an order of magnitude higher than the traditional CVX-based algorithm. The elapsed time of single trajectory planning is stable between 5 ms and 8 ms.

The MPC framework is introduced to the closed-loop simulation under the given disturbance, and the simulation results show that the proposed algorithm can achieve accurate landing under various disturbances. The horizontal position error is less than 0.06 m, the horizontal speed error is less than 0.4 m/s, and the vertical velocity is less than 1.7 m/s.

We noticed that although the DNN-LCvx algorithm has a significant improvement in real-time performance, it also has the following shortcomings that need to be highlighted:

1. In order to obtain a high-precision DNN optimal predictor, the proposed algorithm requires a large number of training samples (1.2 million in this paper); thus, the offline training cost is high.
2. For different types of vehicles, the DNN optimal predictor needs to be retrained, reducing the algorithm's generality.
3. The initial state of the vehicle must be distributed within a specific range (see (18)). Otherwise, the predicted optimal landing time may be invalid.

In summary, the proposed guidance algorithm meets the requirements of multi-constraint, optimality, real-time, and high precision. Moreover, it can significantly improve real-time performance compared with other CVX-based algorithms. Thus, the algorithm can be applied to onboard applications. Some shortcomings of the algorithm are highlighted, which are worth further study in the future.

**Author Contributions:** Conceptualization, W.L. and S.G.; methodology, W.L. and S.G.; software, W.L.; validation, W.L.; formal analysis, W.L.; investigation, W.L.; resources, W.L.; data curation, W.L.; writing—original draft preparation, W.L.; writing—review and editing, W.L. and S.G.; visualization, W.L.; supervision, S.G.; project administration, S.G.; funding acquisition, S.G. All authors have read and agreed to the published version of the manuscript.

**Funding:** This work was supported by the National Natural Science Foundation of China, grant number 11822205 and 11772167.

**Institutional Review Board Statement:** Not applicable.

**Informed Consent Statement:** Not applicable.

**Data Availability Statement:** Not applicable.

**Conflicts of Interest:** The authors declare no conflict of interest.

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
