# Peer review of "Free Final-Time Fuel-Optimal Powered Landing Guidance Algorithm Combing Lossless Convex Optimization with Deep Neural Network Predictor"

_applsci, doi:10.3390/app12073383_

Round 1
Reviewer 1 Report
Review: applsci-1636786 Title: Free Final-time Fuel-optimal Powered Landing Guidance Algorithm Combing Lossless Convex Optimization with DNN PredictorThe paper proposes a real-time approach for powered landing guidance. The suggested control approach is a lossless convex optimization algorithm based on the deep neural network predictor. The proposed method is compared with an iterative convex optimization algorithm. In a closed-loop framework, the proposed approach is combined with a model predictive controller.
The description of the proposed approach and the MPC controller appears to be well designed and the results appear to be coherent. I obviously did not redo their work. I generally feel the paper is interesting.
Generally, the manuscript is well-written and the language level is good. The complete controller design is highly appreciated. However, some comments are as follows:
1- DNN is not defined in the first appearance, I also recommend avoiding the use of it in the title.
2- The argument that the authors draw that the proposed approach is faster than the traditional iterative optimization approaches, ignores the fact that the DNN predictor requires a large amount of training data. The cost of training data is also huge if it relies on experimental data. This needs to be highlighted in the conclusion.
3- In line 387, "is" should be corrected to are.
4- The units in the tables in general should not be after a slash.
5- In line 427. the title of section 5, "Open" should be corrected to Closed.
Reviewer 2 Report
In this paper the Authors present an efficient guidance algorithm for fuel-optimal powered landing.
The manuscript is interesting, fits well with the aim of the Special Issue "Intelligence Sense, Optimization, and Control in Space Vehicles", and it is opinion of the reviewer that it can be published after the following minor revisions.
(1) On page 3 a red writing appears. If this is the Authors' choice, and if so, they must give an explanation, otherwise the reviewer recommends writing it in black.
(2) In Figures 6 and 19 the axis headings (ry/m and rz/m) are graphically unclear. It is advisable to improve the graphics, possibly by rotating the text
According to what said above, the reviewer’s opinion is that the manuscript can be accepted for publication after the described minor revisions.
